# META-LEARNING NEURAL PROCEDURAL BIASES

## ABSTRACT

The goal of few-shot learning is to generalize and achieve high performance on new unseen learning tasks, where each task has only a limited number of examples available. Gradient-based meta-learning attempts to address this challenging task by learning how to learn new tasks by embedding inductive biases informed by prior learning experiences into the components of the learning algorithm. In this work, we build upon prior research and propose *Neural Procedural Bias Meta-Learning* (NPBML), a novel framework designed to meta-learn task-adaptive procedural biases. Our approach aims to consolidate recent advancements in meta-learned initializations, optimizers, and loss functions by learning them simultaneously and making them adapt to each individual task to maximize the strength of the learned inductive biases. This imbues each learning task with a unique set of procedural biases which is specifically designed and selected to attain strong learning performance in only a few gradient steps. The experimental results show that by meta-learning the procedural biases of a neural network, we can induce strong inductive biases towards a distribution of learning tasks, enabling robust learning performance across many well-established few-shot learning benchmarks.

## 1 INTRODUCTION

Humans have an exceptional ability to learn new tasks from only a few example instances. We can often quickly adapt to new domains effectively by building upon and utilizing past experiences of related tasks, leveraging only a small amount of information about the target domain. The field of meta-learning (Schmidhuber, 1987; Vanschoren, 2018; Peng, 2020; Hospedales et al., 2022) explores how deep learning techniques, which often require thousands or even millions of observations to achieve competitive performance, can acquire such a capability. In meta-learning, the learning process is often framed as a bilevel optimization problem (Bard, 2013; Maclaurin et al., 2015; Grefenstette et al., 2019; Lorraine et al., 2020). The outer optimization aims to learn the underlying regularities across a distribution of related tasks and embed them into the inductive biases of a learning algorithm. Consequently, in the inner optimization, the learning algorithm is utilized to quickly adapt to new learning tasks using only a few example instances.

Model-Agnostic Meta-Learning (MAML) (Finn et al., 2017) and its variants (Nichol & Schulman, 2018; Rajeswaran et al., 2019; Song et al., 2019; Triantafillou et al., 2020), are a popular approach to meta-learning. In MAML, the outer optimization aims to learn the underlying regularities across a set of related tasks and embed them into a shared parameter initialization. This initialization is then used in the inner optimization's learning algorithm to encourage fast adaptation to new tasks. While successful, these methods resort to simple gradient descent using the cross-entropy loss for classification or squared loss for regression for the inner learning algorithm. Consequently, subsequent research has extended MAML to meta-learn additional components, such as the learning rate (Behl et al., 2019; Baik et al., 2020), gradient-based optimizer (Li et al., 2017; Lee & Choi, 2018; Simon et al., 2020; Flennerhag et al., 2020; Kang et al., 2023), loss function (Antoniou & Storkey, 2019), and more (Antoniou et al., 2019; Baik et al., 2023). This enables the meta-learning algorithm to induce stronger inductive biases on the learning algorithm, further enhancing performance.

In this paper, we propose *Neural Procedural Bias Meta-Learning* (NPBML), a novel gradient-based framework for meta-learning task-adaptive procedural biases for deep neural networks. Procedural biases are the subset of inductive biases that determine the order of traversal over the search space (Gordon & Desjardins, 1995), they play a central determining role in the convergence, sample efficiency, and generalization of a learning algorithm. As we will show, the procedural biases are

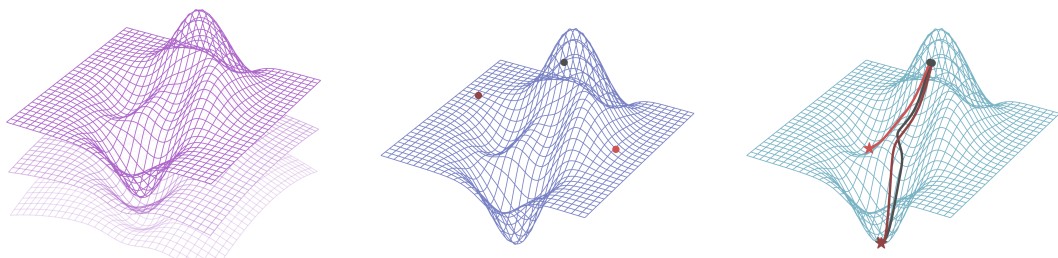

Figure 1: In NPBML, the procedural biases of a deep neural network are meta-learned. This involves meta-learning three key components: the loss function (left), the parameter initialization (center), and the optimizer (right). By meta-learning these components, a strong inductive bias towards fast adaptation can be induced into the learning algorithm.

primarily encoded into three fundamental components of a learning algorithm: the loss function, the optimizer, and the parameter initialization. These components define the geometry of the loss landscape, determine the starting point in this space, and guide the optimization process towards the optimum, respectively, as visualized in Figure 1. Therefore, we aim to meta-learn these three components to maximize the learning performance when using only a few gradient steps.

To achieve this ambitious goal, we first consolidate three related research areas into one unified end-to-end framework: MAML-based learned initializations (Finn et al., 2017), preconditioned gradient descent methods (Lee & Choi, 2018; Flennerhag et al., 2020; Kang et al., 2023), and meta-learned loss functions (Antoniou & Storkey, 2019; Baik et al., 2021; Bechtle et al., 2021; Raymond et al., 2023a;b). We then demonstrate how these meta-learned components can be made task-adaptive through feature-wise linear modulation (FiLM) (Perez et al., 2018) to facilitate downstream task-specific specialization towards each task. The proposed framework is highly flexible and general. As we will show, many existing gradient-based meta-learning approaches arise as special cases of NPBML. To validate the effectiveness of NPBML, we empirically evaluate our proposed algorithm on four well-established few-shot learning benchmarks. The results show that NPBML consistently outperforms many state-of-the-art gradient-based meta-learning algorithms.

## 2 BACKGROUND

In few-shot meta-learning, we are given access to a collection of tasks $\{\mathcal{T}_1, \mathcal{T}_2, \dots\}$, otherwise known as a meta-dataset, where each task is assumed to be drawn from a task distribution $p(\mathcal{T})$. Each task $\mathcal{T}_i$ contains a support set $\mathcal{D}^S$, and a query set $\mathcal{D}^Q$ (*i.e.* a training set and a testing set), where $\mathcal{D}^S \cap \mathcal{D}^Q = \emptyset$. Each of these sets contains a set of input-output pairs $\{(x_1, y_1), (x_2, y_2), \dots\}$. Let $x \in X$ and $y \in Y$ denote the inputs and outputs, respectively. In few-shot meta-learning, the goal is to learn a model of the form $f_\theta(x) : X \to Y$, where $\theta$ are the model parameters. The primary challenge of few-shot learning is that $f_\theta$ must be able to quickly adapt to any new task $\mathcal{T}_i \sim p(\mathcal{T})$ given only a very limited number of instances. For example, in an $N$-way $K$-shot few-shot classification task, $f_\theta$ is only given access to $K$ labeled examples of $N$ distinct classes.

### 2.1 MODEL AGNOSTIC META-LEARNING

MAML (Finn et al., 2017) is a highly influential and seminal method for gradient-based few-shot meta-learning. In MAML, the outer learning objective aims to meta-learn a shared parameter initialization $\boldsymbol{\theta}$ over a distribution of related tasks $p(\mathcal{T})$. This shared initialization embeds prior knowledge learned from past learning experiences into the learning algorithm such that when a new unseen task is sampled fast adaptation can occur. The outer optimization prototypically occurs by minimizing the sum of the final losses $\sum_{\mathcal{T}_i \sim p(\mathcal{T})} \mathcal{L}^{meta}$ on the query set at the end of each task's learning trajectory using gradient descent as follows:

$$\boldsymbol{\theta}_{new} = \boldsymbol{\theta} - \eta \nabla_{\boldsymbol{\theta}} \sum_{\mathcal{T}_i \sim p(\mathcal{T})} \left[ \mathcal{L}^{meta}\Big(\mathcal{D}_i^Q, \theta_{i,j}(\boldsymbol{\theta})\Big) \right] \tag{1}$$

where $\theta_{i,j}(\boldsymbol{\theta})$ refers to the adapted model parameters on the $i^{th}$ task at the $j^{th}$ iterations of the inner update rule $\mathrm{U}^{MAML}$, and $\eta$ is the meta learning rate. The inner optimization for each task starts at the meta-learned parameter initialization $\boldsymbol{\theta}$ and applies $\mathrm{U}^{MAML}$ to the parameters $J$ times:

$$\theta_{i,j}(\boldsymbol{\theta}) = \boldsymbol{\theta} - \alpha \sum_{j=0}^{J-1} \left[ \mathrm{U}^{MAML}\Big(\mathcal{T}_i, \theta_{i,j}(\boldsymbol{\theta})\Big) \right]. \tag{2}$$

In the original version of MAML, the inner update rule $\mathrm{U}^{MAML}$ resorts to simple Stochastic Gradient Descent (SGD) minimizing the loss $\mathcal{L}^{base}$, typically set to the cross-entropy or squared loss, with a fixed learning rate $\alpha$ across all tasks

$$\mathrm{U}^{MAML}\Big(\mathcal{T}_i, \theta_{i,j}(\boldsymbol{\theta})\Big) := \theta_{i,j} - \alpha \nabla_{\theta_{i,j}} \left[ \mathcal{L}^{base}\Big(\mathcal{D}_i^S, \theta_{i,j}\Big) \right]. \tag{3}$$

This approach assumes that all tasks in $p(\mathcal{T})$ should use the same fixed learning rule $\mathrm{U}$ in the inner optimization. Consequently, this greatly limits the performance that can be achieved when taking only a small number of gradient steps with few labeled samples available.

## 3 NEURAL PROCEDURAL BIAS META-LEARNING

In this work, we propose *Neural Procedural Bias Meta-Learning* (NPBML), a novel framework that replaces the fixed inner update rule in MAML, *i.e.*, Equation (3), with a meta-learned task-adaptive learning rule. This modifies the inner update rule in three key ways:

1. An optimizer is meta-learned by leveraging the paradigm of preconditioned gradient descent. This involves meta-learning a parameterized preconditioning matrix $P_{\boldsymbol{\omega}}$ with meta-parameters $\boldsymbol{\omega}$ to warp the gradients of SGD.

2. The conventional loss function $\mathcal{L}^{base}$ (*i.e.* the standard cross-entropy or squared loss) used in the inner optimization is replaced with a meta-learned loss function $\mathcal{M}_{\phi}$, where $\phi$ are the meta-parameters.

3. The meta-learned initialization, optimizer, and loss function are adapted to each new task using Feature-Wise Linear Modulation $FiLM_{\psi}$, a general-purpose preconditioning method that has learnable meta-parameters $\psi$.

For clarity, we provide a high-level overview of NPBML and contrast it to MAML, before expanding on each of the new components in Sections 3.2, 3.3, and 3.4, respectively. Additionally, pseudocode for the outer and inner optimizations is provided in Algorithm 1 and 2, respectively, in Appendix A.

### 3.1 OVERVIEW

The central goal of NPBML is to meta-learn a task-adaptive parameter initialization, optimizer, and loss function. This changes the outer optimization previously seen in Equation (1) to the following, where $\boldsymbol{\Phi} = \{\boldsymbol{\theta}, \boldsymbol{\omega}, \boldsymbol{\phi}, \boldsymbol{\psi}\}$ refers to the set of meta-parameters:

$$\boldsymbol{\Phi}_{new} = \boldsymbol{\Phi} - \eta \nabla_{\Phi} \sum_{\mathcal{T}_i \sim p(\mathcal{T})} \left[ \mathcal{L}^{meta}\Big(\mathcal{D}_i^Q, \theta_{i,j}(\boldsymbol{\Phi})\Big) \right]. \tag{4}$$

Unlike MAML which employs a fixed update rule $\mathrm{U}^{MAML}$ for all tasks, simple SGD using $\mathcal{L}^{base}$, NPBML uses a fully meta-learned update rule

$$\theta_{i,j}(\boldsymbol{\Phi}) = \boldsymbol{\theta}(\boldsymbol{\psi}) - \alpha \sum_{j=0}^{J-1} \left[ \mathrm{U}^{NPBML}\Big(\mathcal{T}_i, \theta_{i,j}(\boldsymbol{\Phi})\Big) \right] \tag{5}$$

which adjusts $\theta_{i,j}$ in the direction of the negative gradient of a meta-learned loss function $\mathcal{M}_{\phi}$. Additionally, the gradient is warped via a meta-learned preconditioning matrix $P_{\boldsymbol{\omega}}$ as follows:

$$\mathrm{U}^{NPBML}\Big(\mathcal{T}_i, \theta_{i,j}(\boldsymbol{\Phi})\Big) := \theta_{i,j} - \alpha P_{(\boldsymbol{\omega},\boldsymbol{\psi})} \nabla_{\theta_{i,j}} \left[ \mathcal{M}_{(\phi,\psi)}\Big(\mathcal{D}_i^S, \theta_{i,j}\Big) \right] \tag{6}$$

where both $\mathcal{M}_{\phi}$ and $P_{\boldsymbol{\omega}}$ are adapted to each task using $FiLM_{\psi}$. This new task-adaptive learning rule empowers each task with a unique set of procedural biases enabling strong and robust learning performance on new unseen tasks in $p(\mathcal{T})$ using only a few gradient steps as depicted in Figure 2.

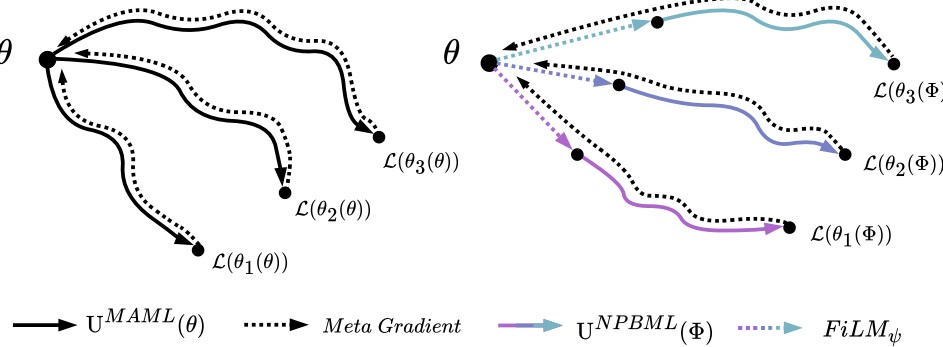

Figure 2: In MAML, the update rule $\mathrm{U}^{MAML}$ optimizes the base model parameters from a shared initialization using simple SGD minimizing $\mathcal{L}^{base}$. In contrast, NPBML adapts the model parameters from a task-adapted initialization using $\mathrm{U}^{NPBML}$, a task-adaptive update rule employing a meta-learned preconditioning matrix $P_{\boldsymbol{\omega}}$ and loss function $\mathcal{M}_{\phi}$.

## 3.2 META-LEARNED OPTIMIZER

In NPBML, the paradigm of Preconditioned Gradient Descent (PGD) (Li et al., 2017; Lee & Choi, 2018; Park & Oliva, 2019; Flennerhag et al., 2020; Simon et al., 2020; Kang et al., 2023) is employed to meta-learn a gradient preconditioner $P_{\boldsymbol{\omega}}$ that rescales the geometry of the parameter space by modifying the gradient descent update rule as follows:

$$\theta_{new} = \theta - \alpha P_{\boldsymbol{\omega}} \nabla_{\theta} \mathcal{M}_{(\boldsymbol{\phi},\boldsymbol{\psi})}. \tag{7}$$

In this work, we take inspiration from T-Nets (Lee & Choi, 2018) and insert linear projection layers $\boldsymbol{\omega}$ into our model $f_{(\boldsymbol{\theta},\boldsymbol{\omega},\boldsymbol{\psi})}$. The model, composed of an encoder $z$ and a classification head $h$,

$$f_{(\boldsymbol{\theta},\boldsymbol{\omega},\boldsymbol{\psi})} = z_{(\boldsymbol{\theta},\boldsymbol{\omega},\boldsymbol{\psi})} \circ h_{\theta} \tag{8}$$

is interleaved with linear projection layers $\boldsymbol{\omega}$ between each layer in the $L$ layer encoder (where $\sigma$ refers to the non-linear activation functions):

$$z_{(\boldsymbol{\theta},\boldsymbol{\omega},\boldsymbol{\psi})}(x_i) = \sigma^{(L)}(\boldsymbol{\theta}^{(L)}\boldsymbol{\omega}^{(L)}(\dots\sigma^{(1)}(\boldsymbol{\omega}^{(1)}\boldsymbol{\theta}^{(1)}x)\dots)). \tag{9}$$

As described in Equations (4)–(6), $\boldsymbol{\omega}$ is meta-learned in the outer loop and held fixed in the inner loop such that preconditioning of the gradients occurs. This form of precondition defines $P_{\boldsymbol{\omega}}$ as a block-diagonal matrix, where each block is defined by the expression $(\boldsymbol{\omega}\boldsymbol{\omega}^{\mathsf{T}})$, as shown in (Lee & Choi, 2018). For a simple model where $f_{(\theta,\boldsymbol{\omega})}(x) = \boldsymbol{\omega}\theta x$, the update rule becomes:

$$\theta_{new} = \theta - \alpha(\boldsymbol{\omega}\boldsymbol{\omega}^{\mathsf{T}})\nabla_{\theta}\mathcal{M}_{(\boldsymbol{\phi},\boldsymbol{\psi})}. \tag{10}$$

We leverage this style of parameterization for $P_{\boldsymbol{\omega}}$ due to its relative simplicity and high expressive power. However, we emphasize that the NPBML framework is highly general, and other forms of preconditioning, such as those presented in (Lee & Choi, 2018; Park & Oliva, 2019; Flennerhag et al., 2020; Simon et al., 2020), could be used instead.

## 3.3 META-LEARNED LOSS FUNCTION

Unlike MAML, which defines the inner loss function $\mathcal{L}^{base}$ to be the cross-entropy or squared loss for all tasks, NPBML uses a meta-learned loss function $\mathcal{M}_{(\phi,\psi)}$ that is learned in the outer optimization. In contrast to handcrafted loss functions, which typically consider only the ground truth label $y$ and the model predictions $f_{(\theta,\boldsymbol{\omega},\boldsymbol{\psi})}(x)$, meta-learned loss functions can, in principle, be conditioned on any task-related information (Bechtle et al., 2021; Baik et al., 2021; Raymond, 2024). In NPBML, the meta-learned loss function is conditioned on three distinct sources of task-related information, which are subsequently processed by three small feed-forward neural networks:

$$\mathcal{M}_{(\phi,\psi)} = \mathcal{L}^{S}_{(\phi,\psi)} + \mathcal{L}^{Q}_{(\phi,\psi)} + \mathcal{R}_{(\phi,\psi)}. \tag{11}$$

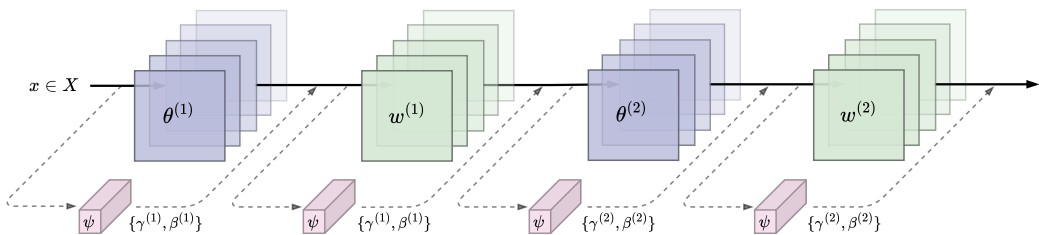

Figure 3: An example of a two-layer convolutional neural network in NPBML, where layers $\theta^{(1)}$ and $\theta^{(2)}$, are interleaved with warp preconditioning layers $\boldsymbol{\omega}^{(1)}$ and $\boldsymbol{\omega}^{(2)}$. Both types of layers are modulated in the inner loop using feature-wise linear modulation layers to induce task adaptation.

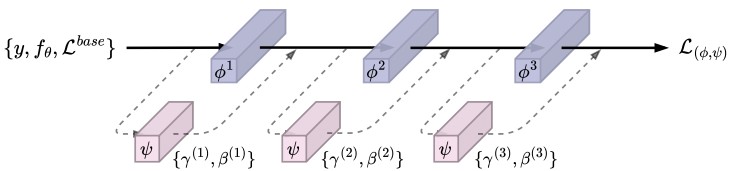

Figure 4: An example of a meta-learned loss function in NPBML, represented as a composition of feed-forward (linear) layers. These layers are modulated using feature-wise linear modulation, resulting in a task-adaptive meta-learned loss function.

First, $\mathcal{L}^S : \mathbb{R}^{2N+1} \to \mathbb{R}^1$ is an inductive loss function conditioned on task-related information derived from the support set; namely, the one-hot encoded ground truth target and model predictions, and the corresponding loss calculated using $\mathcal{L}^{base}$. Next, $\mathcal{L}^Q : \mathbb{R}^{2N+1} \to \mathbb{R}^1$ is a transductive loss function conditioned on task-related information derived from the query set. Here, we give $\mathcal{L}^Q$ access to the model predictions on the query set, embeddings (*i.e.*, relation scores) from a pre-trained relation network (Sung et al., 2018), and the corresponding loss between the model predictions and embeddings using $\mathcal{L}^{base}$. Note that similar embedding functions have previously been used in (Rusu et al., 2019; Antoniou & Storkey, 2019). Finally, the adapted model parameters $\theta_{i,j}$ are used as inputs to meta-learn a weight regularizer $\mathcal{R} : \mathbb{R}^{4L} \to \mathbb{R}^1$. To improve efficiency, we condition $\mathcal{R}$ on the mean, standard deviation, L1, and L2 norm of each layer's weights, as opposed to $\theta_{i,j}$ directly.

## 3.4 TASK-ADAPTIVE MODULATION

Although all tasks in few-shot learning are assumed to be sampled from the same task distribution $p(\mathcal{T})$, the optimal parameter initialization, optimizer, and loss function may differ between tasks. Therefore, in NPBML the meta-learned components $\boldsymbol{\Phi}$ are modulated for each new task, providing each task with a unique set of task-adaptive procedural biases. To achieve this, Feature-wise Linear Modulation (FiLM) layers (Perez et al., 2018; Dumoulin et al., 2018) are inserted into both the encoder $z_{(\boldsymbol{\theta},\boldsymbol{\omega},\psi)}$ and the meta-learned loss function $M_{(\boldsymbol{\phi},\psi)}$ as shown in Figures 3 and 4. $FiLM_\psi$ is defined as follows, where $\gamma$ and $\beta$ are the scaling and shifting vectors, respectively, and $\psi$ are the meta-learnable FiLM parameters:

$$FiLM_\psi(x) = (\gamma_\psi(x) + \mathbf{1}) \odot x + \beta_\psi(x). \tag{12}$$

Affine transformations conditioned on some input have become increasingly popular, being used by several few-shot learning works to make the learned component adaptive (Oreshkin et al., 2018; Jiang et al., 2018; Vuorio et al., 2019; Zintgraf et al., 2019; Baik et al., 2021; 2023). Furthermore, FiLM layers can help alleviate issues related to batch normalization (Ioffe & Szegedy, 2015), which have been empirically observed to cause training instability due to different distributions of features being passed through the same model in few-shot learning (De Vries et al., 2017; Antoniou et al., 2019). In our work, we have found that conditioning the FiLM on the output activations of the previous layers is an effective way to achieve task adaptability. This form of conditioning is in essence a simplified version of that used in CNAPs (Requeima et al., 2019); however, we have omitted the use of global embeddings as we found it was not necessary for our method.

### 3.5 INITIALIZATION

Due to the large number of learnable meta-parameters, initialization becomes an important and necessary aspect to consider. Here we detail how to initialize each of the meta-learned components, *i.e.*, $\Phi_0 = \{\boldsymbol{\theta_0}, \boldsymbol{\omega_0}, \boldsymbol{\phi_0}, \boldsymbol{\psi_0}\}$, in NPBML. Firstly, we pre-train the encoder weights $\boldsymbol{\theta}_0$ prior to meta-learning, following many recent methods in few-shot learning (Rusu et al., 2019; Qiao et al., 2018; Requeima et al., 2019; Ye et al., 2020; Ye & Chao, 2021), see Appendix A.3 for more details. For the linear projection layers $\boldsymbol{\omega}$, we leverage the fact that in PGD, setting $P_{\boldsymbol{\omega}}$ to the identity $\boldsymbol{I}$ recovers SGD. Therefore, we set $\forall l \in \{1, \dots, L\} : \boldsymbol{\omega}^{(l)} = \boldsymbol{I}$; note for convolutional layers this corresponds to Dirac initialization. Regarding the meta-learned loss function $\mathcal{M}_{(\boldsymbol{\phi}, \boldsymbol{\psi})}$, the weights $\boldsymbol{\phi}$ at the start of meta-training are randomly initialized $\boldsymbol{\phi}_0 \sim \mathcal{N}(0, 1e^{-2})$; therefore, the $\mathbb{E}[\mathcal{M}_{(\boldsymbol{\phi}_0, \boldsymbol{\psi}_0)}] = 0$, assuming an identity output activation. Consequently, the definition of the meta-learned loss function in Equation (11) can be modified to

$$\mathcal{M}_{(\boldsymbol{\phi}, \boldsymbol{\psi})} = \mathcal{L}^{base} + \mathcal{L}^{S}_{(\boldsymbol{\phi}, \boldsymbol{\psi})} + \mathcal{L}^{Q}_{(\boldsymbol{\phi}, \boldsymbol{\psi})} + \mathcal{R}_{(\boldsymbol{\phi}, \boldsymbol{\psi})} \tag{13}$$

such that the meta-learned loss function approximately recovers the base loss function at the start of meta-training, *i.e.*, $\mathcal{M}_{(\boldsymbol{\phi}_0, \boldsymbol{\psi}_0)} \approx \mathcal{L}^{base}$. Finally, the FiLM layers in NPBML are initialized using a similar strategy taking advantage of the fact that when $\boldsymbol{\psi}_0 \sim \mathcal{N}(0, 1e - 2)$ the $\mathbb{E}[\gamma_{\boldsymbol{\psi}_0}(x)] = \mathbb{E}[\beta_{\boldsymbol{\psi}_0}(x)] = 0$; consequently, $FiLM_{\boldsymbol{\psi}_0}(x) \approx x$. When initialized in this manner, the update rule for NPBML at the start of meta-training closely approximates the update rule of MAML.

$$\mathrm{U}^{MAML}\Big(\mathcal{T}_i, \theta_{i,j}(\boldsymbol{\theta_0})\Big) \approx \mathbb{E}\left[\mathrm{U}^{NPBML}\Big(\mathcal{T}_i, \theta_{i,j}(\boldsymbol{\Phi_0})\Big)\right] \tag{14}$$

## 4 IMPLICIT META-LEARNING

In NPBML, the parameter initialization, gradient-based optimizer, and loss function are explicitly meta-learned in the outer loop. Critically, we make a novel observation that many other key procedural biases are also implicitly learned by meta-learning these three fundamental components (hence the name given to our algorithm). For example, consider the scalar learning rate $\alpha$, which is implicitly meta-learned since the following equality holds:

$$\exists \alpha \exists \boldsymbol{\phi} : \theta_{i,j} - \alpha \nabla_{\theta_{i,j}} \mathcal{L}^{base} \approx \theta_{i,j} - \nabla_{\theta_{i,j}} \mathcal{M}_{\boldsymbol{\phi}}, \tag{15}$$

and if $\boldsymbol{\phi}$ is made to adapt on each inner step as done in (Baik et al., 2021; Raymond et al., 2023b), then by extension NPBML also learns a learning rate schedule. Another straightforward related observation is that NPBML implicitly learns a layer-wise learning rate $\{\alpha^{(1)}, \dots, \alpha^{(L)}\}$, since for each block $\{\boldsymbol{\omega}^{(1)}, \dots, \boldsymbol{\omega}^{(L)}\}$ in the block diagonal preconditioning matrix $P_{\boldsymbol{\omega}}$ the following holds:

$$\forall l (\exists \boldsymbol{\omega}^{(l)} \exists \alpha^{(l)}) : (\boldsymbol{\omega}^{(l)} (\boldsymbol{\omega}^{(l)})^{\mathsf{T}}) \nabla_{\theta^{(l)}} \mathcal{M}_{\boldsymbol{\phi}} \approx \alpha^{(l)} \nabla_{\theta^{(l)}} \mathcal{M}_{\boldsymbol{\phi}}. \tag{16}$$

There are also less obvious connections that can be drawn. For example, through a transitive relationship, NPBML implicitly learns early stopping when the implicitly learned learning rate approaches zero, as discussed in (Baydin et al., 2018). Furthermore, since there is a linear scaling rule between the batch size and the learning rate (Smith et al., 2017; Smith & Le, 2017; Goyal et al., 2017), NPBML implicitly learns the regularization behavior of the batch size hyperparameter. Another non-trivial example is label smoothing regularization (Müller et al., 2019), which, as proven in (Gonzalez & Miikkulainen, 2020), can be implicitly induced when meta-learning a loss function.

## 5 RELATED WORK

Meta-learning approaches to few-shot learning aim to equip models with the ability to quickly adapt to new tasks given only a limited number of examples by leveraging prior learning experiences across a distribution of related tasks. These approaches are commonly partitioned into three categories: (1) metric-based methods, which aim to learn a similarity metric for efficient class differentiation (Koch et al., 2015; Vinyals et al., 2016; Sung et al., 2018; Snell et al., 2017); (2) memory-based methods, which utilize architectures that store training examples in memory or directly encode fast adaptation algorithms in the model weights (Santoro et al., 2016; Ravi & Larochelle, 2017); and (3) optimization-based methods, which aim to learn an optimization algorithm specifically designed for

fast adaptation and few-shot learning (Finn et al., 2017). This work explored the latter approach by meta-learning a gradient update rule.

MAML (Finn et al., 2017), a highly flexible task and model-agnostic method for meta-learning a parameter initialization from which fast adaptation can occur. Many follow-up works have sought to enhance MAML's performance by addressing limitations in the outer-optimization algorithm, such as the memory and compute efficiency (Nichol & Schulman, 2018; Rajeswaran et al., 2019; Raghu et al., 2019; Oh et al., 2020), or the meta-level overfitting (Rusu et al., 2019; Flennerhag et al., 2018). Relatively fewer works have focused on enhancing the inner-optimization update rule, as is done in our work. Most MAML variants continue to use an inner update rule consisting of SGD with a fixed learning rate minimizing a loss function such as the cross-entropy or squared loss.

Of the works that have explored improving the inner update rule, the vast majority focus on improving the optimizer. For example, early MAML-based methods such as (Behl et al., 2019; Antoniou et al., 2019; Li et al., 2017) explored meta-learning the scalar, layer-wise, and parameter-wise learning rates, respectively. More recent methods have explored more powerful parameterization for the meta-learned optimizer through the utilization of preconditioned gradient descent methods (Lee & Choi, 2018; Park & Oliva, 2019; Flennerhag et al., 2020; Simon et al., 2020; Kang et al., 2023), which rescale the geometry of the parameter space by modifying the update rule with a learned preconditioning matrix. While these methods have advanced MAML-based few-shot learning, they often lack a task-adaptive property, falsely assuming that all tasks should use the same optimizer.

A small number of recent works have also investigated replacing the inner loss function (e.g., cross-entropy loss) with a meta-learned loss function. In (Antoniou & Storkey, 2019), a fully transductive loss function represented as a dilated convolutional neural network is meta-learned. Meanwhile, in (Baik et al., 2021), a set of loss functions and task adapters are meta-learned for each step taken in the inner optimization. Although these approaches have shown a lot of promise, their potential has yet to be fully realized, as they have not yet been meta-learned in tandem with the optimizer as we have done in this work.

## 6 EXPERIMENTAL EVALUATION

In this section, we evaluate the performance of the proposed method on a set of well-established few-shot learning benchmarks. The experimental evaluation aims to answer the following key questions: (1) Can NPBML perform well across a diverse range of few-shot learning tasks? (2) Do the novel components meta-learned in NPBML individually enhance performance? (3) To what extent does each component synergistically contribute to the overall performance of the proposed algorithm?

### 6.1 RESULTS AND ANALYSIS

To evaluate the performance of NPBML, experiments are performed on four well-established few-shot learning datasets: *mini*-Imagenet (Ravi & Larochelle, 2017), *tiered*-ImageNet (Ren et al., 2018), CIFAR-FS (Bertinetto et al., 2018), and FC-100 (Oreshkin et al., 2018). For each dataset, experiments are performed using both 5-way 1-shot and 5-way 5-shot configurations. Results are also reported on both the 4-CONV (Finn et al., 2017; Zintgraf et al., 2019; Flennerhag et al., 2020; Kang et al., 2023) and ResNet-12 (He et al., 2016; Baik et al., 2020; 2021) network architectures. The full details of all experiments, including a comprehensive description of all datasets, models, and training hyperparameters, can be found in Appendix A. The code for our experiments can be found at `https://github.com/*redacted*`

### 6.1.1 MINI-IMAGENET AND TIERED-IMAGENET

We first assess the performance of NPBML and compare it to a range of MAML-based few-shot learning methods on two popular ImageNet derivatives (Deng et al., 2009): *mini*-ImageNet (Ravi & Larochelle, 2017) and *tiered*-ImageNet (Ren et al., 2018). The results, presented in Table 1, demonstrate that the proposed method NPBML, which uses a fully meta-learned update rule in the inner optimization, significantly improves upon the performance of MAML-based few-shot learning methods. The proposed method achieves higher meta-testing accuracy in the 1-shot and 5-shot settings using both low-capacity (4-CONV) and high-capacity (ResNet-12) models.

Table 1: Few-shot classification meta-testing accuracy on 5-way 1-shot and 5-way 5-shot *mini-ImageNet* and *tiered-ImageNet* where ± represents the 95% confidence intervals.

| Method | Base Learner | *mini*-ImageNet (5-way) | | *tiered*-ImageNet (5-way) | |
|---|---|---|---|---|---|
| | | 1-shot | 5-shot | 1-shot | 5-shot |
| MAML[1] | 4-CONV | 48.70±1.84% | 63.11±0.92% | 50.98±0.26% | 66.25±0.19% |
| MetaSGD[2] | 4-CONV | 50.47±1.87% | 64.03±0.94% | - | - |
| T-Net[3] | 4-CONV | 50.86±1.82% | - | - | - |
| MAML++[4] | 4-CONV | 52.15±0.26% | 68.32±0.44% | - | - |
| SCA[5] | 4-CONV | 54.84±0.99% | 71.85±0.53% | - | - |
| WarpGrad[7] | 4-CONV | 52.30±0.80% | 68.40±0.60% | 57.20±0.90% | 74.10±0.70% |
| ModGrad[8] | 4-CONV | 53.20±0.86% | 69.17±0.69% | - | - |
| MeTAL[9] | 4-CONV | 52.63±0.37% | 70.52±0.29% | 54.34±0.31% | 70.40±0.21% |
| ALFA[10] | 4-CONV | 50.58±0.51% | 69.12±0.47% | 53.16±0.49% | 70.54±0.46% |
| GAP[11] | 4-CONV | 54.86±0.85% | 71.55±0.61% | 57.60±0.93% | 74.90±0.68% |
| NPBML | 4-CONV | **57.49±0.83%** | **75.01±0.64%** | **64.24±0.97%** | **79.17±0.71%** |
| MAML[1] | ResNet-12 | 58.60±0.42% | 69.54±0.38% | 59.82±0.41% | 73.17±0.32% |
| MC[6] | WRN-28-10 | - | - | 64.40±0.10% | 80.21±0.10% |
| ModGrad[8] | WRN-28-10 | - | - | 65.72±0.21% | 81.17±0.20% |
| MeTAL[9] | ResNet-12 | 59.64±0.38% | 76.20±0.19% | 63.89±0.43% | 80.14±0.40% |
| ALFA[10] | ResNet-12 | 59.74±0.49% | 77.96±0.41% | 64.62±0.49% | 82.48±0.38% |
| NPBML | ResNet-12 | **61.59±0.80%** | **78.18±0.60%** | **72.22±0.96%** | **85.41±0.61%** |

[1] (Finn et al., 2017)  [2] (Li et al., 2017)  [3] (Lee & Choi, 2018)  [4] (Antoniou et al., 2019)  [5] (Antoniou & Storkey, 2019)  [6] (Park & Oliva, 2019)  [7] (Flennerhag et al., 2020)  [8] (Simon et al., 2020)  [9] (Baik et al., 2021)  [10] (Baik et al., 2023)  11(Kang et al., 2023)

In contrast to PGD methods Meta-SGD, T-Net, WarpGrad, ModGrad, ALFA, and GAP, which meta-learn an optimizer alongside the parameter initialization, NPBML shows clear gains in generalization performance. This improvement is also evident when compared to SCA and MeTAL, which replace the inner optimization's loss function with a meta-learned loss function. These results empirically demonstrate that meta-learning an optimizer and loss function are complementary and orthogonal approaches to improving MAML-based few-shot learning methods.

On *tiered*-ImageNet, the larger of the two datasets, we find that the difference between NPBML and its competitors is even more pronounced than on *mini*-ImageNet. This result suggests that when given enough data, NPBML can learn highly expressive inner update rules that significantly enhances few-shot learning performance. However, meta-overfitting can occur on smaller datasets, necessitating regularization techniques as discussed in Appendix A. Alternatively, we conjecture that less expressive representations for $P_\omega$ would also reduce meta-overfitting.

### 6.1.2 CIFAR-FS AND FC-100

Next, we further validate the effectiveness of NPBML on two popular CIFAR-100 derivatives (Krizhevsky & Hinton, 2009): CIFAR-FS (Ravi & Larochelle, 2017) and FC-100 (Ren et al., 2018). The results, presented in Table 2, show that NPBML continues to achieve strong and robust generalization performance across all settings and models. These results are particularly impressive, given that both MeTAL and ALFA ensemble the top 5 performing models from the same run, which significantly increases the model size and capacity. These experimental results reinforce our claim that meta-learning a task-adaptive update rule is an effective approach to improving the performance of MAML-based few-shot learning algorithms.

Table 2: Few-shot classification meta-testing accuracy on 5-way 1-shot and 5-way 5-shot *CIFAR-FS* and *FC-100* where $\pm$ represents the 95% confidence intervals.

| Method | Base Learner | CIFAR-FS (5-way) | | FC-100 (5-way) | |
|--------|--------------|------------------|------------------|------------------|------------------|
| | | 1-shot | 5-shot | 1-shot | 5-shot |
| MAML[1] | 4-CONV | 57.63±0.73% | 73.95±0.84% | 35.89±0.72% | 49.31±0.47% |
| BOIL[2] | 4-CONV | 58.03±0.43% | 73.61±0.32% | 38.93±0.45% | 51.66±0.32% |
| MeTAL[3] | 4-CONV | 59.16±0.56% | 74.62±0.42% | 37.46±0.39% | 51.34±0.25% |
| ALFA[4] | 4-CONV | 59.96±0.49% | 76.79±0.42% | 37.99±0.48% | 53.01±0.49% |
| NPBML | 4-CONV | **64.90±0.94%** | **79.24±0.69%** | **40.56±0.76%** | **53.48±0.68%** |
| MAML[1] | ResNet-12 | 63.81±0.54% | 77.07±0.42% | 37.29±0.40% | 50.70±0.35% |
| MeTAL[3] | ResNet-12 | 67.97±0.47% | 82.17±0.38% | 39.98±0.39% | 53.85±0.36% |
| ALFA[4] | ResNet-12 | 66.79±0.47% | 83.62±0.37% | 41.46±0.49% | 55.82±0.50% |
| NPBML | ResNet-12 | **69.30±0.91%** | **83.72±0.64%** | **43.63±0.71%** | **59.85±0.70%** |

[1] (Finn et al., 2017)    [2] (Oh et al., 2020)    [3] (Baik et al., 2021)    [4] (Baik et al., 2023)

## 6.2 ABLATION STUDIES

To further investigate the performance of the proposed method, we conduct two sets of ablation studies to analyze the effectiveness of each component. All ablation experiments are performed using the 4-CONV network architecture in a 5-way 5-shot setting on the *mini*-ImageNet dataset.

### 6.2.1 META-LEARNED COMPONENTS

First, we examine the importance of the meta-learned optimizer $P_\omega$, loss function $\mathcal{M}_\phi$, and task-adaptive conditioning method $FILM_\psi$. The results are presented in Table 3, and they demonstrate that each of the proposed components clearly and significantly contributes to the performance of NPBML. In (2) MAML is modified to include gradient preconditioning, which increases accuracy by 2.09%. Conversely in (3) we modify MAML with our meta-learned loss function, resulting in a 6.37% performance increase. Interestingly, the meta-learned loss function enhances performance by a larger margin; however, this may be due to the relatively simple T-Net style optimizer used in NPBML. This suggests that a more powerful parameterization, such as (Flennerhag et al., 2018) or (Kang et al., 2023), may further improve performance. In (4), MAML is modified to include both the optimizer and loss function, resulting in a 7.41% performance increase. This further supports our claim that meta-learning both an optimizer and a loss function are complementary and orthogonal approaches to improving MAML. Finally, in (5), we add our task-adaptive conditioning method, increasing performance by 2.22% over the prior experiment and 9.63% over MAML.

### 6.2.2 META-LEARNED LOSS FUNCTION

The prior ablation study shows that the meta-learned loss function $\mathcal{M}_\phi$ is a crucial component in NPBML. Therefore, we further investigate each of the components; namely, the meta-learned inductive and transductive loss functions, and weight regularizer. The results are presented in Table 4, and surprisingly, they show that each of the components in isolation (7), (8), and (9), improves performance by approximately 5%. However, when combined in (10), the total performance increase is 6.37%. We hypothesize that this result is a consequence of the implicit meta-learning of the learning rate identified in Equation (15), which not only holds for $\mathcal{M}_\phi$, but also for each of its components, *i.e.*, the equality is also true when $\mathcal{M}_\phi$ is replaced with $\mathcal{L}_\phi^S$, $\mathcal{L}_\phi^Q$, or $\mathcal{R}_\phi$. Since all components share implicit learning rate tuning, the performance gains from this behavior do not accumulate; however, the improvement in (10) is better than each component in isolation indicating that each component provides additional unique benefits to the meta-learning process.

Table 3: Ablation study of the meta-learned components in NPBML, reporting the meta-testing accuracy on *mini-ImageNet* 5-way 5-shot. A ✓ denotes that the component is meta-learned, with variant (1) reducing to MAML, while variant (5) represents our final proposed algorithm.

|  | Initialization | Optimizer | Loss Function | Task-Adaptive | Accuracy |
|---|---|---|---|---|---|
| (1) | ✓ | | | | 65.38±0.67% |
| (2) | ✓ | ✓ | | | 67.47±0.68% |
| (3) | ✓ | | ✓ | | 71.75±0.69% |
| (4) | ✓ | ✓ | ✓ | | 72.79±0.67% |
| (5) | ✓ | ✓ | ✓ | ✓ | **75.01±0.64%** |

Table 4: Ablation study of the meta-learned loss function $\mathcal{M}_\phi$ in NPBML, reporting the meta-testing accuracy on *mini-ImageNet* 5-way 5-shot. Note, variants (6) and (10) correspond to variants (1) and (3), respectively in Table 3.

|  | Base Loss | Inductive Loss | Transductive Loss | Weight Regularizer | Accuracy |
|---|---|---|---|---|---|
| (6) | ✓ | | | | 65.38±0.67% |
| (7) | ✓ | ✓ | | | 70.68±0.66% |
| (8) | ✓ | | ✓ | | 70.92±0.68% |
| (9) | ✓ | | | ✓ | 70.04±0.65% |
| (10) | ✓ | ✓ | ✓ | ✓ | **71.75±0.69%** |

## 7 CONCLUSION

In this work, we propose a novel meta-learning framework for learning the procedural biases of a deep neural network. The proposed technique, *Neural Procedural Bias Meta-Learning* (NPBML), consolidates recent advancements in MAML-based few-shot learning methods by replacing the fixed inner update rule with a fully meta-learned update rule. This is achieved by meta-learning a task-adaptive loss function, optimizer, and parameter initialization. The experimental results confirm the effectiveness and scalability of the proposed approach, demonstrating strong few-shot learning performance across a range of popular benchmarks. We believe NPBML provides a principled framework for advancing general-purpose meta-learning in deep neural networks. Looking ahead, numerous compelling future research directions exist, such as developing more powerful parameterizations for the meta-learned optimizer or loss function. We expect that further investigation of this topic will result in more expressive inner update rules, resulting in increased robustness and efficiency within the context of optimization-based meta-learning. Finally, broadening the scope of the proposed framework to encompass the related domains of cross-domain few-shot learning and continual learning may be a promising avenue for future exploration.

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

---

**Algorithm 1** Meta-Learning (Outer Loop)

---

**Input:** $\mathcal{L}^{meta} \leftarrow$ Meta loss function
**Input:** $p(\mathcal{T}) \leftarrow$ Task distribution
**Input:** $\eta \leftarrow$ Meta learning rate

---

1: $\mathbf{\Phi_0} \leftarrow$ Initialize meta-parameters $\{\boldsymbol{\theta}, \boldsymbol{\omega}, \boldsymbol{\phi}, \boldsymbol{\psi}\}$
2: **for** $t \in \{0, ..., \mathcal{S}^{meta}\}$ **do**
3: $\quad \mathcal{T}_0, \mathcal{T}_1, \ldots, \mathcal{T}_B \leftarrow$ Sample tasks from $p(\mathcal{T})$
4: $\quad$ **for** $i \in \{0, ..., B\}$ **do**
5: $\quad\quad \mathcal{D}_i^S = \{(x_i^s, y_i^s)\}_{s=0}^S \leftarrow$ Sample support from $\mathcal{T}_i$
6: $\quad\quad \mathcal{D}_i^Q = \{(x_i^q, y_i^q)\}_{q=0}^Q \leftarrow$ Sample query from $\mathcal{T}_i$
7: $\quad\quad \theta_{i,j} \leftarrow$ Base-Learning using Algorithm (2)
8: $\quad \mathbf{\Phi_{t+1}} \leftarrow \mathbf{\Phi_t} - \eta \frac{1}{B} \nabla_{\mathbf{\Phi_t}} \sum_i \mathcal{L}_i^{meta}(\mathcal{D}_i^Q, \theta_{i,j})$
9: **return** $\mathbf{\Phi_t}$

---

**Algorithm 2** Base-Learning (Inner Loop)

---

**Input:** $\mathcal{L}^{base} \leftarrow$ Base loss function
**Input:** $\mathcal{D}_i^S, \mathcal{D}_i^Q \leftarrow$ Support and query sets
**Input:** $\mathbf{\Phi} \leftarrow$ Meta parameters $\{\boldsymbol{\theta}, \boldsymbol{\omega}, \boldsymbol{\phi}, \boldsymbol{\psi}\}$
**Input:** $\alpha \leftarrow$ Base learning rate
**Input:** $g \leftarrow$ Relation network

---

1: $\theta_{i,0} \leftarrow$ Initialize base weights with $\boldsymbol{\theta}$
2: **for** $j \in \{0, ..., \mathcal{S}^{base}\}$ **do**
3: $\quad \hat{y}_i^S, \hat{y}_i^Q \leftarrow f_{(\theta_{i,j}, \boldsymbol{\omega}, \boldsymbol{\psi})}(x_i^S \cup x_i^Q)$
4: $\quad \mathcal{L}_{i,j}^{base} \leftarrow \frac{1}{|\mathcal{D}^S|} \sum \mathcal{L}^{base}(y_i^S, \hat{y}_i^S)$
5: $\quad \mathcal{L}_{i,j}^S \leftarrow \frac{1}{|\mathcal{D}^S|} \sum \mathcal{L}_{(\boldsymbol{\phi}, \boldsymbol{\psi})}^S(y_i^S, \hat{y}_i^S)$
6: $\quad \mathcal{L}_{i,j}^Q \leftarrow \frac{1}{|\mathcal{D}^Q|} \sum \mathcal{L}_{(\boldsymbol{\phi}, \boldsymbol{\psi})}^Q(g(x_i^Q), \hat{y}_i^Q)$
7: $\quad \mathcal{R}_{i,j} \leftarrow \mathcal{R}_{(\boldsymbol{\phi}, \boldsymbol{\psi})}(\theta_{i,j})$
8: $\quad \mathcal{M}_{(\boldsymbol{\phi}, \boldsymbol{\psi})} \leftarrow \mathcal{L}_{i,j}^{base} + \mathcal{L}_{i,j}^S + \mathcal{L}_{i,j}^Q + \mathcal{R}_{i,j}$
9: $\quad \theta_{i,j+1} \leftarrow \theta_{i,j} - \alpha P \nabla_{\theta_{i,j}} \mathcal{M}_{(\boldsymbol{\phi}, \boldsymbol{\psi})}$
10: **return** $\theta_{i,j}$

---

# A EXPERIMENTAL SETUP

In this section, we provide a comprehensive description of the experimental settings used in this work. Section A.1 offers an overview of the datasets utilized, Section A.2 discusses the network architectures, and Section A.3 details the hyperparameters specific to our proposed method, NPBML. For further details, please refer to Algorithms 1 and 2, as well as our code made available at: `https://github.com/*redacted*`

## A.1 DATASET CONFIGURATIONS

The few-shot learning experiments follow a prototypical episodic learning setup (Ravi & Larochelle, 2017), where each dataset contains a set of non-overlapping meta-training, meta-validation, and meta-testing tasks. Each task $\mathcal{T}_i$ has a support $\mathcal{D}^S$ and query $\mathcal{D}^Q$ set (*i.e.*, training and testing set, respectively). For an $N$-way $K$-shot classification task, the support set contains $N \times K$ instances, and the query set contains $N \times 15$ instances, where $N$ refers to the number of randomly sampled classes, and $K$ refers to the number of instances (*i.e.*, shots) available from each of those classes. For example, in a 5-way 5-shot classification task, the support set contains $5 \times 5 = |\mathcal{D}^S|$ instances to train on, and the query set contains $5 \times 15 = |\mathcal{D}^Q|$ instances to validate the performance on.

### A.1.1 IMAGENET DATASETS

Two commonly used datasets for few-shot learning are two ImageNet derivatives (Deng et al., 2009): *mini*-ImageNet (Ravi & Larochelle, 2017) and *tiered*-ImageNet (Ren et al., 2018). Both datasets are composed of three subsets (training, validation, and testing), each of which consists of images with a size of $84 \times 84$. The two datasets differ in regard to how the classes are partitioned into mutually exclusive subsets.

***mini*-ImageNet** randomly samples 100 classes from the 1000 base classes in ImageNet. Following (Vinyals et al., 2016) the sampled classes are partitioned such that 64 classes are allocated for meta-training, 16 for meta-validation, and 20 for meta-testing, where for each class 600 images are available.

***tiered*-ImageNet** as described in (Ren et al., 2018) alternatively stratifies 608 classes into 34 higher-level categories in the ImageNet human-curated hierarchy. The 34 classes are partitioned into 20 categories for meta-training, 6 for meta-validation, and 8 for meta-testing. Each class in *tiered*-ImageNet has a minimum of 732 instances and a maximum of 1300.

### A.1.2 CIFAR-100 DATASETS

Additional experiments are also conducted on CIFAR-FS (Bertinetto et al., 2018) and FC-100 (Oreshkin et al., 2018), which are few-shot learning datasets derived from the popular CIFAR-100 dataset (Krizhevsky & Hinton, 2009). The CIFAR100 dataset contains 100 classes containing 600 images each, each with a resolution of $32 \times 32$.

**CIFAR-FS** uses a similar sampling procedure to *mini*-ImageNet (Ravi & Larochelle, 2017), CIFAR-FS is derived by randomly sampling 100 classes from the 100 base classes in CIFAR100. The sampled classes are partitioned such that 60 classes are allocated for meta-training, 20 for meta-validation, and 20 for meta-testing.

**FC-100** is obtained by using a dataset construction process similar to *tiered*-ImageNet (Ren et al., 2018), in which class hierarchies are used to partition the original dataset to simulate more challenging few-shot learning scenarios. In FC100 there are a total of 20 high-level classes, where 12 classes are allocated for meta-training, 4 for meta-validation, and 4 for meta-testing.

### A.2 NETWORK ARCHITECTURES

**4-CONV:** Following the encoder architecture settings in (Zintgraf et al., 2019; Flennerhag et al., 2020; Kang et al., 2023), the network consists of four modules. Each module contains a $3 \times 3$ convolutional layer with 128 filters, followed by a batch normalization layer, a ReLU non-linearity, and a $2 \times 2$ max-pooling downsampling layer. Following the four modules, we apply average pooling and flatten the embedding to a size of 128.

**ResNet-12:** The encoder $z_\theta$ follows the standard network architecture settings used in (Baik et al., 2020; 2021). The network, similar to 4-CONV, consists of four modules. Each module contains a stack of three $3 \times 3$ convolutional layers, where each layer is followed by batch normalization and a leaky ReLU non-linearity. A skip convolutional over the convolutional stack is used in each module. Each skip connection contains a $1 \times 1$ convolutional layer followed by batch normalization. Following the convolutional stack and skip connection, a leaky ReLU non-linearity and $2 \times 2$ max-pooling downsampling layer are placed at the end of each module. The number of filters used in each module is $[64, 128, 256, 512]$, respectively. Following the four modules, we apply average pooling and flatten the embedding to a size of 512.

**Classifier:** As MAML and its variants are sensitive to label permutation during meta-testing, we opt to use the permutation invariant classification head proposed by (Ye & Chao, 2021). This replaces the typical classification head $h_\theta \in \mathbb{R}^{in \times N}$ with a *single-weight vector* $\theta_{head} \in \mathbb{R}^{in \times 1}$ which is meta-learned at meta-training time. During meta-testing the weight vector is duplicated into each output, *i.e.*, $h_\theta = \{\theta_c = \theta_{head}\}_{c=1}^N$, and adapted in an identical fashion to traditional MAML. This reduces the number of parameters in the classification head and makes NPBML permutation invariant similar to MetaOptNet (Lee et al., 2019), CNAPs (Requeima et al., 2019), and ProtoMAML (Triantafillou et al., 2020).

**Loss Network:** The meta-learned loss function $\mathcal{M}$ is composed of three separate networks $\mathcal{L}^S$, $\mathcal{L}^Q$, and $\mathcal{R}$ whose scalar outputs are summed to produce the final output as shown in Equation (11). The network architecture for all three networks is identical except for the input dimension of the first layer as discussed in Section 3.3. The network architecture is taken from (Bechtle et al., 2021; Psaros et al., 2022; Raymond et al., 2023b) and is a small feedforward rectified linear neural network with two hidden layers and 40 hidden units in each layer. Note $\mathcal{L}^S$ and $\mathcal{L}^Q$ are applied instance-wise and reduced using a mean reduction, such that they are both invariant to the number of instances made available at meta-testing time in the support and query set.

**Relation Network:** The transductive loss $L^Q$ takes an embedding from a pre-trained relation network (Sung et al., 2018) as one of its inputs, similar to (Rusu et al., 2019; Antoniou & Storkey, 2019). The relation network used in our experiments employs the previously mentioned ResNet-12 encoder to generate a set of embeddings for each instance in $D^S \cup D^Q$. These embeddings are then concatenated and processed using a relation module. The relation module in our experiments consists of two ResNet blocks with $2 \times 512$ and $512$ filters, respectively. The output is then flattened and averaged pooled to an embedding size of $512$, which is passed through a final linear layer of size $512 \times 1$ which outputs each relation score. In regard to training, the relation network is trained using the settings described in (Sung et al., 2018).

### A.3 Proposed Method Settings

#### A.3.1 Pre-training Settings

In NPBML, the encoder portion of the network $z_\theta$ is pre-trained prior to meta-learning, following many recent methods in few-shot learning (Rusu et al., 2019; Qiao et al., 2018; Requeima et al., 2019; Ye et al., 2020; Ye & Chao, 2021). For both the 4-CONV and ResNet-12 models, we follow the pre-training pipeline used in (Ye et al., 2020; Ye & Chao, 2021). During pre-training, we append the feature encoder $z_\theta$ with a temporary classification head $f_\theta$ and train it to classify all classes in the $\mathcal{D}^{train}$ (e.g., over all 64 classes in the *mini*-ImageNet) using the cross-entropy loss. During pre-training, the model is trained over 200000 gradient steps using SGD with Nesterov momentum and weight decay with a batch size of 128. The learning rate, momentum coefficient, and weight decay penalty are set to 0.01, 0.9, and 0.0005, respectively. Additionally, we use a multistep learning rate schedule, decaying the learning rate by a factor of 0.1 at the following gradient steps $\{100000, 150000, 175000, 190000\}$. Note that in our ResNet-12 experiments on *mini*-ImageNet, CIFAR-FS, and FC-100, we observed some meta-level overfitting, which we hypothesize is due to the expressive network architecture coupled with the relatively small datasets. We found that increasing the pre-training weight decay to 0.01 in these experiments resolved the issue.

#### A.3.2 Meta-Learning Settings

In both the meta-training and meta-testing phases, we adhere to the standard hyperparameter values from the literature (Finn et al., 2017). In the outer loop, our algorithm is trained over $S^{meta} = 30,000$ meta-gradient steps using Adam with a meta learning rate of $\eta = 0.00001$. As with previous works, our models are trained with a meta-batch size of 2 for 5-shot classification and 4 for 1-shot classification. In the inner loop, the models are adapted using $S^{base} = 5$ base-gradient steps using SGD with Nesterov momentum and weight decay. The learning rate, momentum coefficient, and weight decay penalty are set to 0.01, 0.9, and 0.0005 respectively. During meta-testing, the same inner loop hyperparameter settings are employed, and the final model is evaluated over 600 tasks sampled from $D^{testing}$. Notably, unlike prior work (Antoniou et al., 2019; Baik et al., 2020; 2021; 2023), we do not ensemble the top 3 or 5 performing models from the same run, as this significantly increases the number of parameters and expressive power of the final models.

#### A.3.3 NPBML Settings

In our experiments, the backbone encoders $z_\theta$ (*i.e.*, 4-CONV and ResNet-12) are modified in two key ways during meta-learning: (1) with linear projection (warp) layers $\omega$ for preconditioning gradients, and (2) feature-wise linear modulation layers $FiLM_\psi$ for task adaptation. In all our experiments, we modify only the last (*i.e.*, fourth) module with warp and FiLM layers based on the findings from (Raghu et al., 2019), which showed that features near the start of the network are primarily reused and do not require adaptation. Additionally, we freeze all preceding modules in both the inner and

outer loops, which significantly reduces the storage and memory footprint of the proposed method, with no noticeable effect on the performance.

Following the recommendations of Flennerhag et al. (2020), warp layers are inserted after the main convolutional stack and before the residual connection ends in the ResNet modules. Regarding the FiLM layers, they are inserted after each batch normalization layer in a manner similar to the backbone encoders used in (Requeima et al., 2019). Where for convolutional layers, we first pool in the spatial dimension to obtain the average activation scores per map, and then flatten in the depth/filter dimension before processing $FiLM_\psi$ to obtain $\gamma$ and $\beta$. As discussed in Section 3.4, the loss networks $\mathcal{L}^S$, $\mathcal{L}^Q$, and $\mathcal{R}$ also use FiLM layers, these are interleaved between each of the linear layers as shown in Figure 4.

## B  FURTHER DISCUSSION

The proposed NPBML framework is a highly flexible and general approach to gradient-based meta-learning. As a result of this generality, many existing meta-learning algorithms are closely related to NPBML, with some potentially arising as special cases of NPBML. In what follows, we discuss some salient examples:

- **MAML++**: In (Antoniou et al., 2019), the inner optimization of MAML is modified by learning per-step batch normalization running statistics, weights and biases, and layer-wise learning rates. This is closely related to the FiLM layers used in NPBML, as they also output meta-learned scale and shift parameters, *i.e.*, $\gamma_\psi$ and $\beta_\psi$, which improve robustness to varying feature distributions. Furthermore, as shown by Equation (16), NPBML implicitly learns a layer-wise learning rate through $P_\omega$.

- **SCA**: In (Antoniou & Storkey, 2019), the authors propose replacing $\mathcal{L}^{base}$ with a meta-learned loss function $\mathcal{M}_\phi$ similar to NPBML. The primary difference between their parameterization and ours is that their meta-learned loss function is represented as a 1D dilated convolutional neural network with DenseNet-style residual connectivity. In our experiments, we found that a simple feedforward ReLU network is sufficiently expressive to obtain high performance as shown in the ablation study in Table 3.

- **MeTAL**: In (Baik et al., 2021), a set of task-adaptive loss functions, represented as small feedforward networks are meta-learned for each step in the inner optimization. Task-adaptivity is achieved by using a separate set of networks to generate FiLM scale and shift parameters which are applied directly to $\phi$ instead of the network's activations. Unlike MeTAL, NPBML does not maintain separate networks for each step; thus, it can be applied to more gradient steps at meta-testing time than it was initially trained to do.

- **ALFA**: In (Baik et al., 2023), MAML's inner optimization is modified with meta-learned layer-wise learning rate values and weight decay coefficients. As shown in Equation (16), NPBML implicitly learns layer-wise learning rate values. Additionally, since the learned loss function $\mathcal{M}_\phi$ in NPBML is conditioned on $\theta_{i,j}$, our method also implicitly learns a scalar weight decay values, since $\exists\lambda\exists\phi : \theta_{i,j} - \nabla_{\theta_{i,j}}(\mathcal{L}^{base} + \lambda||\theta||^2) \approx \theta_{i,j} - \nabla_{\theta_{i,j}}\mathcal{M}_\phi$.

- **MetaSGD**: In (Li et al., 2017), a parameter-wise learning rate is meta-learned in tandem with the parameter initialization. This corresponds to using an identical inner update rule to Equation (6) where $\mathcal{M}_{(\phi,\psi)} = \mathcal{L}^{base}$ and $P_{(\omega,\psi)} = diag(\omega)$, *i.e.*, no meta-learned loss function, and a diagonal matrix used instead of a block diagonal matrix for gradient preconditioning.

- **WarpGrad**: In (Flennerhag et al., 2020), a full gradient preconditioning matrix is induced by modifying the meta-learned linear projection layers employed in T-Nets (Lee & Choi, 2018) with non-linear activations. Furthermore, a trajectory-agnostic meta-objective is used. As NPBML utilizes T-Net style gradient preconditioning, simply introducing non-linear activation functions and replacing the outer objective $\mathcal{L}^{meta}$ in Equation (4) would be sufficient to recover WarpGrad's gradient preconditioning.

- **ModGrad**: In (Simon et al., 2020), a low-rank gradient preconditioning matrix is meta-learned. This is, in essence, an identical inner update rule to Equation (6) where $\mathcal{M}_{(\phi,\psi)} = \mathcal{L}^{base}$ and $P_{(\omega,\psi)} = \omega_1 \otimes \omega_2$, where $\omega_1 \in \mathbb{R}^{in \times r}$, $\omega_2 \in \mathbb{R}^{r \times out}$, and $\otimes$ is the outer product between the two low-rank matrices with rank $r$.

