# OpenReview forum: "Meta-Learning Neural Procedural Biases"
_ICLR.cc/2025/Conference — Submitted to ICLR 2025_

### Official Review · Reviewer_iJPA · 2024-10-30

**Soundness:** 4
**Presentation:** 4
**Contribution:** 2
**Rating:** 3
**Confidence:** 3

**Summary:**

The authors combine together three existing forms of meta-learning: namely, learning intializations, optimizers and loss functions. Through experiments on miniimagenet, tiered-imagenet, and CIFAR, the authors show that the approach outperforms baselines.

**Strengths:**

**Originality**

The paper combines together existing meta-learning techniques in a way that I believe has not been seen in prior literature.

**Quality**

Given that there is no theory introduced in this paper, the experimental results are very important. Fortunately, the experiments seem thoroughly conducted with many baselines and the proposed method strongly outperforms baselines on all experiments. Ablations are also included, which is good practice.

**Clarity**

The paper is quite well written and the figures and tables are well-presented. The form of the ablation tables (3, 4) are a good example for other papers to follow.

**Significance**

The paper will likely be of significance to meta-learning practitioners who are trying to achieve state-of-the-art performance on meta-learning tasks.

**Weaknesses:**

The originality of this work appears limited to me. The authors simply seem to combine existing techniques and find that the performance improves, which is not very surprising. This also limits the significance of the paper to a wider audience. I would encourage the authors to extend the work in at least one direction; some ideas are: 1) including a theoretical result, 2) proposing new performance improvements to the existing methods, 3) demonstrating the emergence of a new phenomenon when multiple meta-learning techniques are combined.

Also, in terms of practical use, it will be very important to compare the computational cost of the proposed method against baselines.

Overall, however, this is a technically solid work that lacks sufficient novely and impact.

**Questions:**

How does the computational cost (runtime, memory) of the proposed method compare to baselines?
Are the results state-of-the-art on the tasks tested?
Can the authors prove a theoretical result about the proposed method?

---

> ### Author Response · Authors · 2024-11-21
>
> Thank you for taking the time to review our paper, and for highlighting the originality, quality, clarity, and significance. In what follows we address your questions:
>
> **RE: Method Novelty**
>
> Prior works has explored extending MAML to meta-learning additional components [1-9]. Many of these methods are meta-learning components that have previously been meta-learned in the literature — however, this does not imply that they are not novel, as the strategies for integrating the components and how they are meta-learned are vital for the downstream performance (e.g., WarpGrad [10] extending T-Net’s [11]), often yielding very different performance.
>
> One of the key novel contributions of this work (Section 4) was to show that by meta-learning three key components, the parameter initialization, optimizer, and loss function, you can implicitly meta-learn other components such as the scalar and parameter-wise learning rate, batch size, weight regularizer and more. Due to this implicit behavior, many existing prior meta-learning algorithms become special case of our proposed method (Appendix B).
>
> **RE: Computational and Memory Complexity**
>
> Compared to the baseline, MAML, our proposed method requires less runtime — on MiniImageNet 5way-5shot using the 4-CONV model the runtime is 11.4 hours compared to MAML’s 15.5 hours (on single A6000 GPU). This reduced runtime is achieved by utilizing a pre-trained backbone to initialize the encoder weights (Appendix A3), thus requiring only half the meta-gradient steps. In addition, MAML’s computational graph can be reused to avoid recomputing trajectory information, thus there is minimal storage of computational overhead when going from MAML to our proposed method. In the final manuscript we will add an additional section in the appendix to further discuss the proposed methods runtime in contrast to the baseline.
>
> **RE: State-Of-The-Art Performance**
>
> The performance of our proposed algorithm is to the best of our knowledge SOTA, or near SOTA on optimization-based few-shot learning methods (under the constraints of the same data resources and models). There are methods such as [12] which have superior performance, however, this is due to them using additional data resources, as well as higher capacity network architectures.
>
> We would like to emphasize to the reviewer that the central goal of this work was not primarily to achieve SOTA performance, it was to show that there are three key components that should be meta-learned in order to learn the procedural biases of a deep neural network.
>
> **RE: Theoretical Results**
>
> Thank you for the suggestion. What additional theoretical results (in addition to those discussed above) does the review recommend to further enhance the contribution of the paper?
>
>
> —
>
> We hope we have managed to clarify the points of confusion and address your concerns. We kindly, ask you to consider updating your score, if you feel we have answered your questions adequately. If there are any further questions, please do not hesitate to reach out during the short rebuttal period.
>
> Best regards,
>
> The Authors
>
> —
>
> [1] Behl, HS, et al. "Alpha MAML: Adaptive Model-Agnostic Meta-Learning." ICML2019.
>
> [2] Baik, S, et al. "Meta-Learning with Adaptive Hyperparameters." NeurIPS2020.
>
> [3] Li, Z., et al. Meta-SGD: Learning to Learn Quickly for Few-Shot Learning. arXiv2017.
>
> [4] Lee, Y., et al. Gradient-Based Meta-Learning with Learned Layerwise Metric and Subspace. ICML2018.
>
> [5] Antoniou, A, et al. How to Train your MAML. ICLR2019.
>
> [6] Antoniou, A. et al. Learning to Learn by Self-Critique. NeurIPS2019.
>
> [7]  Park, E., et al. Meta-Curvature. NeurIPS2019.
>
> [8] Simon, C., et al. On Modulating the Gradient for Meta-Learning. ECCV2020.
>
> [9] Baik, S., et al. Meta-Learning with Task-Adaptive Loss Function for Few-Shot Learning. ICCV2021.
>
> [10] Flennerhag, S., et al. Meta-learning with Warped Gradient Descent. ICLR2020.
>
> [11] Lee, Y., et al. Gradient-Based Meta-Learning with Learned Layerwise Metric and Subspace. ICML2018.
>
> [12] Hu, S, et al. "Pushing the Limits of Simple Pipelines for Few-Shot Learning: External Data and Fine-Tuning make a Difference." CVPR2022.

---

> > ### Comment · Reviewer_iJPA · 2024-11-21
> >
> > Thank you for your thorough response.
> >
> > I'm glad to see the runtime analysis. I still think it would be more impactful to extend the work further in some direction.
> >
> > Regarding a theoretical result, one thing that would be interesting to show is how existing meta-generalization guarantees can be combined to produce a meta-generalization guarantee for the proposed combined method. For instance, suppose meta-parameters $\phi_1$, $\phi_2$, ... $\phi_N$ individually have generalization guarantees of the form:
> >
> > $L_i \leq C_i T^{-p}$
> >
> > with probability $1-\delta$ where $L_i$ is the expected loss on an unseen task when meta-training parameter $\phi_i$ is trained (and other $\phi_j$ are fixed), $C_i$ and $p$ are constants and $T$ is the number of training tasks. Then, can the authors show a generalization guarantee on the expected loss when all the meta-training parameters $\phi_i$ are meta-trained?
> >
> > I would be happy to increase my rating if the authors could show this (or a similar theoretical result), or otherwise a new method or insight.

---

### Official Review · Reviewer_kj58 · 2024-11-01

**Soundness:** 2
**Presentation:** 3
**Contribution:** 2
**Rating:** 3
**Confidence:** 4

**Summary:**

The paper proposes a bilevel optimization meta-learning algorithm that combines meta-learned initializations, meta-learned preconditioners and meta-learned loss functions with task-specific feature-wise linear modulation models.
It shows that this combination achieves competitive performance on visual few-shot classification task compared to other bilevel optimization meta-learning methods.

**Strengths:**

The paper is clearly presented, motivating the combination of existing components into a new algorithm well.

**Weaknesses:**

While the experimental baselines contain a number of bilevel optimization-based meta-learning algorithms that fall into the same paradigm, comparisons to other popular paradigms such as extended pretraining of the backbone (the presented method also pretrains the backbone) and in-context learning / sequence modelling are missing.
Such methods [e.g. 1, 2] achieve stronger performance on the few-shot learning tasks evaluated here albeit using larger models.
In combination with the large computational complexity of bilevel optimization, the significance of the presented method is therefore unclear to me (see questions).

[1] Hu, Shell Xu, et al. "Pushing the limits of simple pipelines for few-shot learning: External data and fine-tuning make a difference." Proceedings of the IEEE/CVF Conference on Computer Vision and Pattern Recognition. 2022.

[2] Fifty, Christopher, et al. "Context-aware meta-learning." arXiv preprint arXiv:2310.10971 (2023).

**Questions:**

- Can you elaborate how you situate your method in comparison to non bilevel optimization methods? My impression is that such methods [e.g. 1, 2] are both cheaper to run and perform better. Do you disagree with this statement? Where do you see the advantages of your method in comparison to pretraining / sequence modeling based approaches?
- The reported meta learning rate of $\eta= 0.00001$ combined with only 30'000 meta-steps seems to be relatively low and makes me wonder how much meta-learning actually happens. One number that would help to contextualize this would be the performance your model achieves at initialization without any meta-training (i.e. setting $\eta= 0.0$ but keeping everything else equal).
- Are the reported baselines reproductions or are the numbers taken from their respective papers?

---

> ### Author Response · Authors · 2024-11-21
>
> Thank you for reviewing our paper, and for the kind words regarding the presentation and clarity. Below we address your questions:
>
> **RE: Comparison to Non-Bi-level Optimization Methods**
>
> Regarding methods such as [1, 2] these methods utilize significantly larger more powerful models (as you noted), as well as they use additional data resources which we (and other optimization-based methods) do not; hence, it is not fair to directly compare their performance.
>
> In regard to your question, there is an inherent tradeoff between these two competing paradigms. Bilevel optimization methods are more computationally expensive at meta-training time; however, since the models are more compact, the models require less memory and compute at inference time when deployed, relative to pre-training methods which rely on scaling up the model and use significantly more data (which is not always available). Therefore, depending on the requirements of the problem and the domain, one technique may be preferable to another; however, pre-training methods do not Pareto dominate optimization-based methods. Thanks for bringing this up, we will include a discussion on this topic in the updated version of the manuscript.
>
> **RE: Performance at Initialization**
>
> To further contextualize the performance before and after meta-training, we ran some experiments on mini-ImageNet using no meta-training and the results are shown below. The results clearly show that meta-training is vital for obtaining competitive few-shot learning performance. We will include these additional results in the appendix of the manuscript.
>
> | Method                  | 5way-1shot (4-CONV) | 5way-5shot (4-CONV) | 5way-1shot (ResNet-12) | 5way-5shot (Resnet-12) |
> | :-------------------- | :------: | ----: | :------: | ----: |
> | NPBML (no meta-training)   |   49.44%   |  63.44%  |   54.89%  |   70.75%  |
> | NPBML (with meta-training) |  **57.49%**    | **75.01%** |  **61.59%** |  **78.18%**|
>
> **RE: Comparison Methods Numbers**
>
> The results reported in Table 1 and 2 are taken from the respective papers. The only exception to this was MAML, where the tiered-ImageNet, CIFAR-FS, and FC-100 results were inherited directly from [3], as the results did not exist in the original MAML paper but were important to include as a point of reference.
>
> —
>
> We hope we have managed to clarify the points of confusion and address your concerns. We kindly, ask you to consider updating your score, if you feel we have answered your questions adequately. If there are any further questions, please do not hesitate to reach out during the short rebuttal period.
>
> Best regards,
>
> The Authors
>
> —
>
> [1] Hu, S, et al. "Pushing the Limits of Simple Pipelines for Few-Shot Learning: External Data and Fine-Tuning make a Difference." CVPR2022.
>
> [2] Fifty, C, et al. "Context-Aware Meta-Learning." ICLR2024.
>
> [3] Baik, S, et al. "Meta-Learning with Task-Adaptive Loss Function for Few-Shot Learning." ICCV2021.

---

> > ### Comment · Reviewer_kj58 · 2024-11-25
> >
> > Thank you for your concise response and providing the additional performance numbers prior to meta-learning. I am assuming these numbers have been obtained after the initial pre-training step, please correct me if this assumption is wrong.
> >
> > > pre-training methods do not Pareto dominate optimization-based methods.
> >
> > After reading the other reviews and your responses, I remain sceptical of the impact of this work given the noticeable performance gap towards methods that do not rely on bilevel optimization style meta-learning such as [1,2] as mentioned in my original review. In particular, if it hinges on the claim that "pre-training methods do not Pareto dominate optimization-based methods" then there need to be respective experiments/analysis/references to back this claim which to my knowledge are currently missing.
> >
> > > Thanks for bringing this up, we will include a discussion on this topic in the updated version of the manuscript.
> >
> > Has this already been included in the manuscript? Could you point me to the corresponding line numbers if so?

---

### Official Review · Reviewer_W9cu · 2024-11-02

**Soundness:** 3
**Presentation:** 3
**Contribution:** 2
**Rating:** 5
**Confidence:** 4

**Summary:**

This work proposes to meta-learn task-adaptive procedural biases by simultaneously learning initializations, optimizers, and loss functions that adapt to each specific task. It demonstrates that by meta-learning these components, the framework NPBML can induce strong inductive biases towards a distribution of learning tasks, leading to robust performance across several few-shot learning benchmarks.

**Strengths:**

1. NPBML combines the gradient-based meta-learning methods into a unified end-to-end framework, which meta-learns the key components of learning, i.e., initializations, optimizers, and loss functions, simultaneously. It enables meta-learning to acquire more optimization components and potentially enhances performance.

2. The framework is flexible and general, with many existing gradient-based meta-learning approaches emerging as special cases within NPBML.

**Weaknesses:**

1. There is a risk of meta-overfitting, where the model learns too well from the meta-training tasks and fails to generalize to new, unseen tasks. Although the authors mention this issue in the paper and suggest that it can be alleviated using regularization techniques, this introduces many manual choices, which contradicts the goal of automatically learning to learn from tasks. How to prevent meta-overfitting within the NPBML framework should be carefully discussed.

2. Although the authors state that the gradient-based optimizer is meta-learned, the number of steps to be updated in the inner loop is still a manually set hyperparameter. The article mentions that the early stopping mechanism can be learned implicitly, but in the experimental setup, 5 steps are used instead of a larger number to leverage this early stopping mechanism.

3. The networks used in the experiments are 4-CONV and ResNet-12. It remains questionable whether this framework is still effective on larger convolutional networks or transformer architectures.

4. The tasks used in the experiments are limited to 5-way 1-shot and 5-way 5-shot classification, which is quite different from the tasks that need to be addressed in real-world scenarios, such as segmentation, detection, super-resolution, translation, text summarization, and so on. The effectiveness of this framework in practical task scenarios has not been validated.

**Questions:**

1. How to prevent meta-overfitting within the NPBML framework?
2. How is the number of update steps in the inner loop determined? What would be the effect of using a large number of update steps, such as 50 or 100, in the inner loop?
3. Is this framework still effective for larger convolutional networks or transformer architectures?
4. What would the experimental results be when using this framework on more realistic tasks, such as segmentation or detection?

Please see Weaknesses for details.

---

> ### Author Response · Authors · 2024-11-21
>
> Thank you for taking the time to review our paper, and for all the feedback and suggestions you have given. We address your questions below:
>
> **RE: Meta-Overfitting and Regularization**
>
> Thank you for raising the topic of meta-overfitting. As shown in Table 1 and 2, our empirical results show that our proposed method attains superior *out-of-sample testing performance* on four well established few-shot learning datasets compared to 11 other few shot learning methods.
>
> Analogous to conventional supervised learning settings, a more powerful and expressive meta-learning technique is exposed to a higher risk of meta-overfitting, however, this can be mitigated through regularization techniques. In meta-learning, meta-regularization techniques such as proximal regularization [12] could be leveraged to avoid over adaptation to new tasks in the inner optimization. In the final manuscript we will include some additional discussion on meta-regularization.
>
> **RE: Inner Loop Hyperparameters**
>
> All hyperparameters where possible were taken from the established literature. Regarding the inner loop gradient steps, this was taken from [1], in order to ensure a fair comparison between methods. As our works primary contribution was not about meta-optimization, simple unrolled differentiation was used identical to MAML. Unrolled differentiation scales linearly in memory with respect to the number of inner gradient steps; however, this can be obviated by using techniques such as implicit differentiation [12] or trajectory agnostic meta-optimization techniques such as [7, 13, 14].
>
> **RE: Larger Models**
>
> The models selected for our experiments were chosen following the experimental protocol from established literature [1-11]. As we do not have access to larger compute resources we cannot evaluate on larger convolutional or vision transformers models. However, the results in Table 1 and 2, indicate that our methods performance scales in a comparable manner to existing methods on both the few-shot learning CIFAR-100 and ImageNet partitions.
>
> **RE: Additional Applications**
>
> Regarding the applications domains explored, our paper performs extensive experiments on a diverse range of few-shot image classification tasks — these being, mini-ImageNet, tiered-ImageNet, CIFAR-FS, and FC-100, which are the most established and recognized datasets in the area. While it would be valuable to explore further applications domains such as: segmentation, detection, super-resolution, translation, text summarization etc., due to space limitations as well as ICLR not being an application focused conference, we leave this for future work to explore.
>
> —
>
> We hope we have managed to clarify the points of confusion and address your concerns. We kindly, ask you to consider updating your score, if you feel we have answered your questions adequately. If there are any further questions, please do not hesitate to reach out during the short rebuttal period.
>
> Best regards,
>
> The Authors
>
> —
>
> [1]  Finn, C., et al. Model-Agnostic Meta-Learning for Fast Adaptation of Deep Networks. ICML2017.
>
> [2] Li, Z., et al. Meta-SGD: Learning to Learn Quickly for Few-Shot Learning. arXiv2017.
>
> [3] Lee, Y., et al. Gradient-Based Meta-Learning with Learned Layerwise Metric and Subspace. ICML2018.
>
> [4] Antoniou, A, et al. How to Train your MAML. ICLR2019.
>
> [5] Antoniou, A. et al. Learning to Learn by Self-Critique. NeurIPS2019.
>
> [6]  Park, E., et al. Meta-Curvature. NeurIPS2019.
>
> [7] Flennerhag, S., et al. Meta-learning with Warped Gradient Descent. ICLR2020.
>
> [8] Simon, C., et al. On Modulating the Gradient for Meta-Learning. ECCV2020.
>
> [9] Baik, S., et al. Meta-Learning with Task-Adaptive Loss Function for Few-Shot Learning. ICCV2021.
>
> [10] Baik, S., et al. Learning to Learn Task-Adaptive Hyperparameters for Few-Shot Learning. TPAMI2023.
>
> [11] Kang, S., et al. Meta-Learning with a Geometry-Adaptive Preconditioner. CVPR2023.
>
> [12] Rajeswaran, A., et al. Meta-Learning with Implicit Gradients. NeurIPS2019.
>
> [13] Flennerhag, S., et al. Transferring Knowledge Across Learning Processes. ICLR2019.
>
> [14] Flennerhag, S., et al. Bootstrapped meta-learning. ICLR2022.

---

### Official Review · Reviewer_hiLK · 2024-11-04

**Soundness:** 3
**Presentation:** 2
**Contribution:** 2
**Rating:** 5
**Confidence:** 4

**Summary:**

The authors propose to combine several meta-learning methods into a single one, which they dub “Neural Procedural Biases Meta-Learning” (NPBML). The main idea is to meta-learning the initialization, optimizer, and loss function of a neural network over a distribution of tasks. They show gains on few-shot image classification benchmarks (FC100, CIFAR-FS, mini-/tiered-ImageNet), and perform ablation studies showing each component adds to the overall performance.

While I think the paper is sound, I find it lacks novelty and significance. While combining existing methods shows promising results on (outdated) benchmarks, the results are not compelling enough to justify acceptance. Moreover, the resulting method is actually similar to the MT-nets of Lee et al., 2018; the major difference seems to be learning FiLM layers instead of binary masks. For these reasons I think the paper should be revised and resubmitted.

**Strengths:**

- While low-hanging, the motivation to combine multiple meta-learning approaches is sound. I mention some caveats below, but I can see how enabling different components of the meta-learner to adapt could accelerate convergence and boost final performance.
- In all their experiments, the authors’ method performs the best. The ablations also support their claims.

**Weaknesses:**

Conceptual weaknesses:

- While it’s tempting to combine existing meta-learning work, a major caveat is not discussed: the more powerful the meta-learner, the higher the risk of meta-overfitting. In other words, the meta-learner risks to overfit to the train task distribution and fail to adapt to new unseen distributions. I wished the authors mentioned this trade-off — and others that arise from designing stronger meta-learner — explicitly and potentially even addressed it directly.
- None of the components in the proposed combination are novel. So this paper is only an incremental contribution, especially since the empirical results are underwhelming (more below). I would also like to note that the final combination is very close to the 8-year old work of Lee et al., 2018 (MT-nets): MT-nets also learn an optimizer, they also learn an initialization, the loss function is implicitly learned by the optimizer in the last network’s layer, and they also learn a modulating function. The main difference seems to be that here the modulating function uses FiLM layers whereas MT-nets uses binary masks.
- The proposed method is more difficult to implement and computationally more expensive than alternatives. This is a significant weakness, which the authors should also mention. For example, what is the runtime of their methods vs MAML, ProtoNet, or SimpleShot?

Experimental weaknesses:

- The benchmarks used in this work are somewhat outdated and don’t challenge modern meta-learning methods. In fact, I believe this is why the proposed method outperforms all other baselines: none of the benchmarks challenge the meta-generalization ability of the methods — instead, they reward overfitting to the train task distribution.
- Additionally, I think some baselines are missing. For example, ProtoNet or SimpleShot mentioned above. One could even make the argument that these two methods deserve larger backbone architectures, given their inference-time adaptation is much faster than gradient-based algorithms like NPBML.

**Questions:**

See my questions in the weaknesses section.

---

> ### Author Response · Authors · 2024-11-21
>
> Thank you for taking the time to review our paper and for highlighting our strong few-shot learning performance and soundness. Below we address your questions:
>
> **RE: Meta-Overfitting**
>
> Meta-overfitting and meta-regularization are important topics of research, similar to conventional learning paradigms. However, we would like to emphasize to the reviewer that it is not the central topic under study in this paper. This paper aims to propose a method for meta-learning task-specific procedural biases which are specifically designed to attain strong learning performance. The experimental results in Table 1 and 2, confirm that our proposed method can attain superior *out-of-sample testing performance* on four well established few-shot learning datasets compared to 11 other few shot learning methods. If our method is found to overfit on other datasets, popular meta-regularization techniques from the literature could be leveraged such as proximal regularization [1].
>
> **RE: Method Novelty**
>
> Prior works has explored extending MAML to meta-learning additional components [2-11]. Many of these methods are meta-learning components that have previously been meta-learned in the literature — however, this does not imply that they are not novel, as the strategies for integrating the components and how they are meta-learned are vital for the downstream performance (e.g. WarpGrad [11] extending T-Net’s [5]), often yielding very different performance.
>
> One of the key novel contributions of this work (Section 4) was to show that by meta-learning three key components, the parameter initialization, optimizer, and loss function, you can implicitly meta-learn other components such as the scalar and parameter-wise learning rate, batch size, weight regularizer and more. Due to this implicit behavior, many existing prior meta-learning algorithms become special case of our proposed method (Appendix B).
>
> Regarding MT-Nets, they do not implicitly learn a loss function as the last layer of the model does not use transformation (T) layers, only the last layer of the encoder does (see https://github.com/yoonholee/MT-net). Furthermore, meta-learning a loss function is not equivalent to meta-learning a T-Layer in the classifier, as this would result in the classification head being frozen in the inner loop resulting in no adaptation of the classifier at meta-testing time. Finally, as shown in Table 1 and the ablation in Table 3, our method shows substantial improvements over MT-Nets, e.g. 51.7% accuracy vs 57.49% on mini-ImageNet 5-way 1-shot.
>
> **RE: Implementation and Computational Complexity**
>
> Concerning the implementation difficulty of our proposed method, NPBML, is not significantly more difficult to implement then MAML, and in many cases it would just be a drop in replacement of a few lines of code (see our attached code). As for the computational complexity, much of the computational burden is obviated by reusing the optimization trajectory store by MAML. Furthermore, as we utilize pre-trained backbones to initialize the encoder weights (Appendix A3), following recent SOTA methods [12-16], our method only requires half the number of meta-training gradient steps. On mini-ImageNet 5way-5shot using the 4CONV model the runtime is 11.4 hours compared to MAML’s 15.5 hours. In the final manuscript we will add an additional section in the appendix to further discuss the runtime of our algorithm.
>
> **RE: Benchmark Datasets**
>
> Thank you for raising this topic. While we are agree that the CIFAR-100 and ImageNet partitions are not the perfect benchmark, they are by a significant margin, the most popular and most well-established dataset used in this area. Therefore, to aid in cross comparison, we performed experiments using these dataset.
>
> **RE: ProtoNet and SimpleShot**
>
> Regarding the benchmarks, we chose the 11 most closely related optimization-based meta-learning methods to compare against in Table 1. In the updated manuscript we will include further experiments comparing to ProtoNet [17] and SimpleShot [18]. As we use a standardized experimental setting we can directly compare the results as follows:
>
> | Method                  | 5way-1shot (4-CONV) | 5way-5shot (4-CONV) |
> | :-------------------- | :------: | ----: |
> | ProtoNet [17]         |   49.42%   | 68.20% |
> | SimpleShot [18]     |   33.17%   | 63.25% |
> | NPBML (Ours)       |  **57.49%**    | **75.01%** |
>
> As shown, our method performs significantly better than both ProtoNet and SimpleShot. Note, although the “inference-time adaptation” may be faster then optimization-based approaches, these methods often rely on significantly larger network architectures in order to get better performance; consequently, this greatly increase their inference costs when making predictions when deployed.

---

> > ### Comment · Reviewer_hiLK · 2024-11-25
> >
> > Thank you for your response, I'll add some comments below.
> >
> > - I remain under the impression that the meta-overfitting issue and the choice of benchmarks conspire to make the method look better than what it is.
> > - In terms of computational complexity, I find it surprising that MAML takes longer than your method given your method is a superset of MAML. Does MAML also use the pretrained backbone? If not, this would explain the runtime and (partly) the lower performance.
> > - For ProtoNet and SimpleShot, you should use a larger backbone to keep computational comparisons apples-to-apples. These 2 methods don't adapt via gradient descent at test-time, so they need more compute in the representation extractor in order to work well.
> >
> > In light of the rebuttal I'll keep my original score.

---

> ### Author Response · Authors · 2024-11-21
>
> —
>
> Apologies for the lengthy response. We hope we have managed to clarify the points of confusion and address your concerns. We kindly, ask you to consider updating your score, if you feel we have answered your questions adequately. If there are any further questions, please do not hesitate to reach out during the short rebuttal period.
>
> Best regards,
>
> The Authors
>
> —
>
> [1] Rajeswaran, A, et al. "Meta-Learning with Implicit Gradients." NeurIPS2019.
>
> [2] Behl, HS, et al. "Alpha MAML: Adaptive Model-Agnostic Meta-Learning." ICML2019.
>
> [3] Baik, S, et al. "Meta-Learning with Adaptive Hyperparameters." NeurIPS2020.
>
> [4] Li, Z., et al. Meta-SGD: Learning to Learn Quickly for Few-Shot Learning. arXiv2017.
>
> [5] Lee, Y., et al. Gradient-Based Meta-Learning with Learned Layerwise Metric and Subspace. ICML2018.
>
> [6] Antoniou, A, et al. How to Train your MAML. ICLR2019.
>
> [7] Antoniou, A. et al. Learning to Learn by Self-Critique. NeurIPS2019.
>
> [8]  Park, E., et al. Meta-Curvature. NeurIPS2019.
>
> [9] Simon, C., et al. On Modulating the Gradient for Meta-Learning. ECCV2020
>
> [10] Baik, S., et al. Meta-Learning with Task-Adaptive Loss Function for Few-Shot Learning. ICCV2021
>
> [11] Flennerhag, S., et al. Meta-learning with Warped Gradient Descent. ICLR2020.
>
> [12] Rusu, A., et al. Meta-Learning with Latent Embedding Optimization. ICLR2019.
>
> [13] Qiao, Siyuan, et al. "Few-Shot Image Recognition by Predicting Parameters from Activations”. CVPR2018.
>
> [14] Requeima, J., et al. "Fast and Flexible Multi-Task Classification using Conditional Neural Adaptive Processes”. NeurIPS2019.
>
> [15] Ye, H., et al. “Few-Shot Learning via Embedding Adaptation with Set-to-Set Functions”. CVPR2020.
>
> [16] Ye, H. J., et al. “How to Train Your MAML to Excel in Few-Shot Classification”. ICLR2022.
>
> [17] Snell, J., et al. Prototypical Networks for Few-Shot Learning. NeurIPS2017.
>
> [18] Wang, Y., et al. SimpleShot: Revisiting Nearest-Neighbor Classification for Few-Shot Learning. arXiv2019.

---

### Meta-Review · Area_Chair_TKpc · 2024-12-20

**Metareview:**

This paper combines gradient-based meta-learning methods for few-shot learning, learning initialisations, optimisers and loss functions together, with empirical improvements upon existing methods.

Reviewers largely agree that it is interesting to combine these methods together, but also agree that the novelty of the paper is low as the methods are the same. I agree with this assessment. Overall, all reviewers agree that this paper is below the acceptance threshold. There are also points raised about the experimental setup (eg by Reviewer hiLK and W9cu: both reviewers did not feel that the author rebuttal sufficiently addressed their points). For what is primarily an empirical paper, addressing these comments in a future version should help.

**Additional Comments On Reviewer Discussion:**

Reviewers mostly felt that their major concerns were not adequately addressed by the authors during the rebuttal, and the authors did not respond to follow-up questions. This applies to concerns by Reviewer hiLK, W9cu and kj58. Please see their responses to author rebuttal.

Reviewer iJPA asked for theoretical results. Such a result is one way to improve the novelty of the paper, but I think is out-of-scope of this paper, which is primarily an empirical one. I agree with the overall comment from iJPA about needing to further the work in some direction (a sentiment all reviewers agreed on).

---

### Decision · Program_Chairs · 2025-01-22

Reject